# Periodontal Condition Is Correlated with Deep and Subcortical White Matter Hyperintensity Lesions in Japanese Adults

**DOI:** 10.3390/ijerph17051694

**Published:** 2020-03-05

**Authors:** Minako Hada, Tetsuji Azuma, Koichiro Irie, Takatoshi Yonenaga, Kazutoshi Watanabe, Fumiko Deguchi, Akihiro Obora, Takao Kojima, Takaaki Tomofuji

**Affiliations:** 1Department of Community Oral Health, School of Dentistry, Asahi University, Mizuho, Gifu 501-0296, Japan; mi_n_a@dent.asahi-u.ac.jp (M.H.); tetsuji@dent.asahi-u.ac.jp (T.A.); ora102@osaka-med.ac.jp (T.Y.); 2Department of Oral Health and Preventive Dentistry, Meikai University School of Dentistry, Sakado, Saitama 350-0283, Japan; coichiro@dent.meikai.ac.jp; 3Asahi University Hospital, 3- 23 Hashimoto-cho, Gifu, Gifu 500-8523, Japan; watanabe@murakami.asahi-u.ac.jp (K.W.); deguchi5757@yahoo.co.jp (F.D.); a-obora@murakami.asahi-u.ac.jp (A.O.); tkojima-gi@umin.ac.jp (T.K.)

**Keywords:** periodontitis, brain, risk, cross-sectional study

## Abstract

Deep and subcortical white matter hyperintensity (DSWMH) lesions are a small-vessel disease of the brain. The aim of this cross-sectional study was to investigate the relationship between DSWMH lesions and periodontal status in Japanese adults who participated in a health check. We enrolled 444 consecutive participants (mean age, 54.5 years) who received both brain and oral health evaluation services at the Asahi University Hospital. Magnetic resonance imaging was used to detect DSWMH lesions. Periodontal status was assessed using the community periodontal index. Of the study participants, 215 (48.4%) had DSWMH lesions. Multivariate logistic regression showed that the presence of DSWMH lesions was significantly related to age ≥ 65 years (vs. < 65 years, odds ratio [OR] = 2.984, 95% confidence interval [CI] = 1.696–5.232), systolic blood pressure ≥ 140 mmHg (vs. < 140 mmHg, OR = 2.579, 95% CI = 1.252–5.314), the presence of ≥ 28 teeth (vs. < 28 teeth, OR = 0.635, 95% CI = 0.420–0.961), and probing pocket depth (PPD) ≥ 6 mm (vs. PPD < 6 mm, OR = 1.948, 95% CI = 1.132–3.354) after adjustment for confounding factors. Having PPD ≥ 6 mm may be a risk factor for DSWMH lesions in Japanese adults.

## 1. Introduction

Periodontitis is a chronic inflammatory disease of the oral cavity, which is initiated by the overgrowth of anaerobic bacteria in periodontal pockets [1]. Progression of periodontitis depends on the interaction between anaerobic bacteria and host, leading to inflammatory molecule production that reflects periodontal tissue destruction. It has been reported that periodontitis is associated with elevated circulating inflammatory molecules, including C-reactive protein (CRP) [2,3]. Elevation of circulating CRP levels is also associated with the risk of numerous systemic diseases, including Alzheimer’s disease (AD) and age-related brain decline [4,5]. Therefore, it is feasible that brain diseases and periodontitis would have a positive association. 

Clinical studies have investigated the relationship between brain diseases and periodontitis. A cross-sectional study indicates that severe periodontitis may be an important independent predictor of early-onset poststroke depression status in patients with acute ischemic stroke [6]. A case-control study shows that periodontitis was significantly associated with the presence of chronic migraine, independent of well-known migraine risk factors [7]. Cohort studies reveal that patients with periodontitis exhibited a higher risk of developing AD [8] and dementia [9,10] than those without periodontitis. A retrospective cohort study reports that patients with periodontitis have a higher risk of ischemic stroke compared to those in the gingivitis cohort [11]. These observations indicate that periodontitis could be associated with brain diseases. However, since there is still little information about the relationship between brain diseases and periodontitis in humans, additional clinical studies are needed.

In Japan, health evaluation services in hospitals, including brain evaluations, are popular. In brain health evaluation services, magnetic resonance imaging (MRI) is used to detect various brain diseases, such as deep and subcortical white matter hyperintensity (DSWMH) lesions [12]. White matter hyperintensity lesions are a subcortical occlusive small-vessel disease, and these changes in white matter are likely to be associated with aging [13], hypertension [14,15,16], and diabetes [15,16]. However, the complete etiology of DSWMH lesions remains unclear. 

Determining the correlation between DSWMH lesions and periodontal status would be useful in understanding the relationship between brain diseases and periodontitis. Here, we hypothesized that the risk of DSWMH lesions might be associated with periodontitis among Japanese adults. The purpose of this cross-sectional study was to investigate the relationship between DSWMH lesions and periodontal status in Japanese adults. 

## 2. Materials and Methods 

### 2.1. Study Population 

Participants for this study consisted of Japanese adults who underwent both brain and oral health evaluations from January 2013 through December 2017 at the Asahi University Hospital in Gifu, Japan. Accordingly, 444 participants (mean age, 54.5 years; 305 men, 139 women) were eligible for this study. The study protocol was approved by the Ethics Committee of Asahi University (No. 27010). All participants provided written informed consent prior to study participation. 

### 2.2. Evaluation of DSWMH Lesions 

MRI examinations were performed on all participants using a superconducting magnet with a main field strength of 1.5 T. An experienced doctor classified DSWMH lesions on MRI imaging (0 = absent; 1 = ≤ 3 mm, small foci and regular margins; 2 = ≥ 3 mm, large foci; 3 = diffusely confluent; 4 = extensive changes in the white matter) according to the Japanese Brain Dock Guidelines 2003 [17].

### 2.3. Evaluation of Oral Condition

Three dentists examined the oral status of the study participants. Periodontal status was assessed using the Community Periodontal Index (CPI) [18]. Ten teeth were selected for periodontal examination: two molars in each posterior sextant, and the upper right and lower left central incisors. Measurements were made using a CPI probe (YDM, Tokyo, Japan) at six sites (mesio-buccal, mid-buccal, disto-buccal, disto-lingual, mid-lingual, and mesio-lingual) per tooth. The presence or absence of teeth exhibiting bleeding on probing (BOP) was recorded. The number of teeth in the mouth was also counted. Good intra- and inter-examiner agreement was achieved for repeated probing pocket depth (PPD) measurements in the 10 teeth used for CPI (Kappa statistic, > 0.8).

### 2.4. Different Variables Related to DSWMH Lesions 

We collected information about age, sex, smoking habit (presence/absence), drinking habit (presence/absence), and regular exercise (presence/absence). Height and body weight were measured using the automatic height scale with body composition meter (TBF-110/TBF-210/DC-250, TANITA, Tokyo, Japan). Body mass index (BMI) was calculated as weight in kilograms divided by the square of height in meters. BMI ≥ 25 was defined as overweight or obese [19]. Systolic blood pressure (SBP) and diastolic blood pressure (DBP) were measured using an automatic blood pressure monitor (HBP-9021/HBP-9020/BP-230RV3, OMRON HEALTHCARE, Kyoto, Japan). Hypertension was defined as SBP ≥ 140 mmHg or DBP ≥ 90 mmHg [20].

Fasting venous blood samples were collected. Serum concentrations of triglyceride, high-density lipoprotein (HDL) cholesterol, low-density lipoprotein (LDL) cholesterol, and C-reactive protein (CRP) were determined using a simultaneous multi-item automatic analyzer (Dimension Vista 1500, Siemens Healthineers Japan, Tokyo, Japan). Serum levels of hemoglobin A1c (HbA1c) were determined using a diabetes automatic analyzer (DM-JACK, Kyowa Medex, Tokyo, Japan). Abnormal serum lipid levels were defined as triglyceride ≥ 150 mg/dL and/or HDL cholesterol < 40 mg/dL [21]. In addition, elevated CRP levels and poor glycemic control were defined as CRP ≥ 0.3 mg/dL [22] and HbA1c ≥ 6.5% [23], respectively.

### 2.5. Statistical Analysis

In this study, one or more teeth with ≥ 4 mm PPD was defined as the presence of moderate periodontitis [24,25]. One or more teeth with ≥ 6 mm PPD was defined as the presence of severe periodontitis [25]. Since the maximum number of present teeth, excluding wisdom teeth, is usually 28, we analyzed the number of present teeth with a cut-off value of 28. We evaluated the normality of our data using the Kolmogorov–Smirnov test. Because all continuous variables were not normally distributed, data are expressed as median (first and third quartiles). The Chi-square test and Mann–Whitney U test were used to assess significant differences in selected characteristics between study participants with and without DSWMH lesions. The Chi-square test was used for three group comparisons with different severity of periodontitis. Univariate and multivariate logistic regression analyses using a stepwise method (backward selection approach) were performed with the presence (grade 1–4) or absence (grade 0) of DSWMH lesions as dependent variables. In this model, the presence or absence of one or more teeth with ≥ 4 mm PPD or ≥ 6 mm PPD were included as the independent variable. Other independent variables were selected when the p-value was < 0.05 in the univariate model. Furthermore, Spearman’s correlation analysis between each variable was performed, and variables with |r| > 0.8 were removed to avoid multicollinearity [23]. 

Analyses were performed using the SPSS statistical package (IBM SPSS statistics version 25, IBM Japan, Tokyo, Japan). All reported p-values were considered statistically significant if less than 0.05. 

## 3. Results

### 3.1. Characteristics of the Participants

The overall prevalence of DSWMH lesions was 48.4%. The prevalence of DSWMH lesions with age < 65 years and < 28 teeth was 34.9% (*n* = 155) and 23.6% (*n* = 104), respectively. Table 1 presents the characteristics of the participants. There were significant differences between the participants with and without DSWMH lesions in age, SBP, DBP, and serum HbA1c level (*p* < 0.001). There were also significant differences between the participants with and without DSWMH lesions in a number of teeth present and prevalence of PPD ≥ 6 mm (*p* < 0.01). 

Results of comparisons of the participants with different severity of periodontitis are shown in Table 2. No participants had grade 4 DSWMH lesions. There were significant differences among the participants with PPD ≤ 3 mm, PPD = 4–5 mm, and PPD ≥ 6 mm in the grade of DSWMH lesions (*p* < 0.05). 

### 3.2. Results of Spearman’s Correlation Analysis between Each Variable

The presence or absence of DSWMH lesions was significantly correlated with age ≥ 65 years, SBP ≥ 140 mmHg, number of present teeth ≥ 28 teeth, PPD ≥ 4 mm, and PPD ≥ 6 mm (*p* < 0.05) (Table 3). There were no variables with |r|  >  0.8 in the Spearman’s correlation analysis. 

### 3.3. Logistic Regression Analysis with Prevalence of DSWMH Lesions as the Dependent Variable

Table 4 and Table 5 present the results of the logistic regression analysis with prevalence of DSWMH lesions as the dependent variable. In univariate logistic regression analysis, the prevalence of DSWMH lesions was related with age ≥ 65 years (vs. < 65 years, odds ratio [OR] = 3.834, 95% confidence interval [CI] = 2.237–6.571), SBP ≥ 140 mmHg (vs. < 140 mmHg, OR = 2.932, 95% CI = 1.460–5.891), number of teeth present ≥ 28 (vs. < 28 teeth, OR = 0.489, 95% CI = 0.333–0.720), presence of PPD ≥ 4 mm (vs. absence, OR = 1.524, 95% CI = 1.014–2.293), and presence of PPD ≥ 6 mm (vs. absence, OR = 2.184, 95% CI = 1.298–3.677). 

In multiple logistic regression analysis, the prevalence of DSWMH lesions was related with age ≥ 65 years (vs. < 65 years, OR = 2.984, 95% CI = 1.696−5.232), SBP ≥ 140 mmHg (vs. < 140 mmHg, OR = 2.579, 95% CI = 1.252−5.314), number of teeth present ≥ 28 (vs. < 28 teeth, OR = 0.635, 95% CI = 0.420−0.961) and presence of PPD ≥ 6 mm (vs. absence, OR = 1.948, 95% CI = 1.132−3.354), after adjusting for age, SBP, number of teeth present, and PPD ≥ 6 mm. In contrast, the prevalence of DSWMH lesions was not related with PPD, after adjusting for age, SBP, number of teeth present, and PPD ≥ 4 mm.

## 4. Discussion

To the best of our knowledge, this is the first study to investigate the relationship between DSWMH lesions and periodontal status. In this study, the grade of DSWMH lesions varied according to the severity of periodontitis. In addition, logistic regression analysis showed that the risk of DSWMH lesions was higher in participants with PPD ≥ 6 mm than without PPD ≥ 6 mm (OR = 1.948; 95% CI = 1.132−3.354). This indicates that severe periodontitis could be a risk factor for DSWMH lesions. In contrast, a significant correlation was not observed between DSWMH lesions and the presence of PPD ≥ 4 mm. It is feasible that severe but not moderate periodontitis influences the risk for DSWMH lesions. 

Clinical studies have investigated the relationship between brain diseases and periodontitis. A cross-sectional study showed that severe periodontitis was an independent predictor for early-onset poststroke depression status [6]. A case-control study demonstrated that mild cognitive impairment, subjective cognitive decline, and AD were associated with increased numbers of deep periodontal pockets (OR = 8.43; 95% CI = 4.00–17.76) [26]. A cohort study reported that patients with 10 years of chronic periodontitis exposure had a higher risk of developing AD than the unexposed groups (adjusted hazard ratio [aHR] = 1.707; 95% CI = 1.152–2.528) [8]. Another cohort study also revealed that participants with chronic periodontitis had an elevated risk for overall dementia (aHR = 1.06; 95% CI = 1.01–1.11) and AD (aHR = 1.05; 95% CI = 1.00–1.11) compared with those without periodontitis [10]. Moreover, a retrospective cohort study indicated that dental care services, including scaling and intensive treatment, reduced the risk of ischemic stroke in patients with periodontitis [11]. These observations are consistent with our findings, suggesting that periodontitis might be a risk factor for brain diseases. 

A variety of mechanisms may underlie the relationship between brain diseases and periodontitis. For instance, it is known that periodontitis can induce an elevation in systemic inflammation [2,3], and such conditions would increase the risk of brain diseases [4,5]. However, our data did not support this hypothesis, because the risk of DSWMH lesions was not associated with elevated CRP levels. In addition, a case-control study found that serum IgG levels to common periodontal microbiota are associated with increased risk of developing incident AD [27]. Animal studies suggest that periodontal pathogen infections in the oral cavity resulted in brain colonization and augmented AD pathogenesis [28,29]. In this study, we did not measure circulating periodontal pathogens. Future studies will need to evaluate circulating periodontal pathogens to clarify the mechanisms responsible for the relationship between periodontitis and DSWMH lesions.

Our observations also indicated that the risk of DSWMH lesions was lower in participants with ≥ 28 teeth, compared with those with < 28 teeth. The quality of oral function increases according to the number of teeth [30], and the presence of ≥ 28 teeth indicates good oral function. Therefore, the risk of DSWMH lesions and the quality of oral function may have a negative association. However, in our analyses, there was no association between the presence of ≥ 20 teeth and the risk of DSWMH lesions. In the present study, the prevalence of participants with < 20 teeth was only 3.4% (data not shown). The association between the presence of ≥ 20 teeth and the risk of DSWMH lesions might not have reached the level of significance, since almost all of the current study participants had ≥ 20 teeth.

Aging [13] and hypertension [14,15,16] are known risk factors for white matter hyperintensity lesions. Our observations also showed that the risk of DSWMH lesions was associated with aging (≥ 65 years) and hypertension (SBP ≥ 140 mmHg). However, although previous studies indicated that diabetes and dyslipidemia are risk factors for white matter lesions [15,16], the risk of DSWMH lesions was not associated with poor glycemic control and abnormal cholesterol levels in our study. Furthermore, there were no correlations between DSWMH lesions and serum lipids, including triglyceride and HDL cholesterol. Thus, the present data suggest that metabolic diseases, except hypertension, had little effect on DSWMH lesions. 

In this study, the prevalence of DSWMH lesions was 48.4%. This value was relatively high compared to previous studies, which reported that the prevalence of white matter hyperintensities was 25.9% in 1249 young clinical outpatients [31] and 26.2% in 141 asymptomatic individuals aged ≥ 50 years [32]. In addition, the prevalence of PPD = 4–5 mm and PPD ≥ 6 mm was 59.2% and 16.4%, respectively (Table 2). In Japanese aged 45–64 years old, the prevalence of PPD = 4–5 mm and PPD ≥ 6 mm was reported as 37% and 21%, respectively [33]. Therefore, the prevalence of periodontitis in the present study appears to be relatively high, while the prevalence of severe periodontitis was similar to previous studies. 

DSWMH lesions are closely associated with the risk of stroke [34,35]. The degree of white matter lesions is also associated with cognitive function decline [36]. These studies indicate that suppressing the risk of DSWMH lesions may be effective in preventing stroke and cognitive function decline. The present study did not investigate the direct effects of severe periodontitis on brain disorders. However, our data support the hypothesis that improving severe periodontitis may suppress the risk of DSWMH lesions, contributing to stroke prevention and reducing cognitive function decline. 

There are some limitations in the present study. First, all participants received health evaluations at Asahi University Hospital. As the participants were regional, this limits the ability to extrapolate the present findings to the general population. Second, no data was collected regarding the occlusal condition and medication status, which may modulate the present association between DSWMH lesions and periodontal status. Third, this was a cross-sectional study, which does not permit conclusions regarding causal relationships. Therefore, prospective cohort studies and interventional studies are required to provide further mechanistic clarification. Finally, it is important to clarify the effects of periodontitis prevention on improvements in brain health. 

## 5. Conclusions

There appears to be a positive association between DSWMH lesions and severe periodontitis in a cross-sectional study in Japan. 

## Figures and Tables

**Table 1 ijerph-17-01694-t001:** Characteristics of study participants with and without DSWMH lesions.

Variables	DSWMH Lesions		*p*-Value ^1^
	Absence (*n* = 229)	Presence (*n* = 215)	
Gender (male, %)	153 (66.8)	152 (70.7)	0.413
Age, years	50 (44, 57)	59 (54, 65)	<0.001
BMI	23.3 (20.6, 25.4)	23.3 (21.3, 25.1)	0.975
Smoking habits (presence, %)	36 (15.7)	32 (14.9)	0.895
Drinking habits (presence, %)	48 (21.0)	57 (26.5)	0.181
Regular exercise (presence, %)	51 (22.3)	42 (19.5)	0.487
SBP, mmHg	119 (109, 127)	126 (116, 134)	<0.001
DBP, mmHg	73 (65, 79)	77 (69, 85)	<0.001
Serum parameters			
HbA1c, %	5.4 (5.2, 5.6)	5.6 (5.3, 5.8)	<0.001
Triglyceride, mg/dL	74 (49, 111)	74 (52, 118)	0.361
HDL cholesterol, mg/dL	61 (50, 74)	59 (51, 76)	0.864
CRP, mg/dL	0.05 (0.02, 0.09)	0.04 (0.03, 0.08)	0.612
Oral parameters			
Number of present teeth	28 (27, 29)	28 (26, 29)	0.005
BOP (presence, %)	195 (85.2)	179 (83.3)	0.604
PPD ≥ 4 mm (presence, %)	149 (65.1)	159 (74.0)	0.050
PPD ≥ 6 mm (presence, %)	26 (11.4)	47 (21.9)	0.003

Continuous variables were expressed as median (first quartile, third quartile). ^1^ Chi-square test or Mann–Whitney U test. Abbreviation: DSWMH, deep and subcortical white matter hyperintensity; BMI, body mass index; SBP, systolic blood pressure; DBP, diastolic blood pressure; HbA1c, hemoglobin A1c; HDL, high-density lipoprotein; LDL, low-density lipoprotein; CRP, C-reactive protein; BOP, bleeding on probing; PPD, probing pocket depth.

**Table 2 ijerph-17-01694-t002:** Differences in DSWMH grade according to periodontal condition.

Grade	PPD ≤ 3 mm	PPD 4–5 mm	PPD ≥ 6 mm
0	80 ^1^	123	26
1	22	58	14
2	27	46	28
3	7	8	5
4	0	0	0

^1^ Number of participants.

**Table 3 ijerph-17-01694-t003:** Spearman’s correlation analysis between each variable.

Variables	DSWMH Lesions ^1^	Gender ^2^	Age ≥ 65 Years ^1^	BMI ≥ 25 ^1^	Smoking Habits ^1^	Drinking Habits ^1^	Regular Exercise ^1^	SBP ≥ 140 mmHg ^1^	DBP ≥ 90 mmHg ^1^	HbA1c ≥ 6.5% ^1^	Triglyceride ≥ 150 mg/dL ^1^	LDL Cholesterol ≥ 120 mg/dL ^1^	HDL Cholesterol < 40 mg/dL ^1^	CRP ≥ 0.3 mg/dL ^1^	Number of Present Teeth ≥ 20 Teeth ^1^	Number of Present Teeth ≥ 28 Teeth ^1^	BOP ^1^	PPD ≥ 4 mm ^1^
Gender ^2^	−0.04	-	-	-	-	-	-	-	-	-	-	-	-	-	-	-	-	-
Age ≥ 65 years ^1^	0.24 *	−0.01	-	-	-	-	-	-	-	-	-	-	-	-	-	-	-	-
BMI ≥ 25 ^1^	−0.02	−0.18 *	−0.05	-	-	-	-	-	-	-	-	-	-	-	-	-	-	-
Smoking habits ^1^	−0.01	−0.17 *	−0.09	0.09	-	-	-	-	-	-	-	-	-	-	-	-	-	-
Drinking habits ^1^	0.07	−0.22 *	−0.02	−0.04	0.15 *	-	-	-	-	-	-	-	-	-	-	-	-	-
Regular exercise ^1^	−0.03	−0.01	0.04	−0.11 *	−0.13 *	−0.03	-	-	-	-	-	-	-	-	-	-	-	-
SBP ≥ 140 mmHg ^1^	0.15 *	−0.15 *	0.11 *	0.06	0.03	0.11 *	0.004	-	-	-	-	-	-	-	-	-	-	-
DBP ≥ 90 mmHg ^1^	0.04	−0.16 *	−0.09	0.05	0.02	0.10 *	0.10 *	0.57 *	-	-	-	-	-	-	-	-	-	-
HbA1c ≥ 6.5% ^1^	0.02	−0.12	−0.01	0.11 *	0.04	0.003	−0.09	0.07	0.03	-	-	-	-	-	-	-	-	-
Triglyceride ≥ 150 mg/dL ^1^	−0.01	−0.18 *	−0.10 *	0.21 *	0.20 *	0.001	−0.06	0.12 *	0.09	0.004	-	-	-	-	-	-	-	-
LDL cholesterol ≥ 120 mg/dL ^1^	−0.01	0.02	−0.05	0.08	−0.01	−0.10 *	−0.06	0.05	0.01	−0.08	0.15 *	-	-	-	-	-	-	-
HDL cholesterol < 40 mg/dL ^1^	−0.03	−0.11	0.01	0.21 *	0.13 *	−0.003	−0.06	−0.02	−0.04	0.05	0.38 *	0.001	-	-	-	-	-	-
CRP ≥ 0.3 mg/dL ^1^	−0.002	−0.08	0.06	0.14 *	0.14 *	0.002	−0.01	0.02	−0.03	0.13 *	0.02	−0.01	0.15 *	-	-	-	-	-
Number of present teeth ≥ 20 teeth ^1^	−0.07	0.02	−0.17 *	−0.01	−0.09 *	−0.07	−0.10 *	−0.07	0.01	−0.05	0.04	−0.04	−0.01	−0.17 *	-	-	-	-
Number of present teeth ≥ 28 teeth ^1^	−0.17 *	−0.02	−0.29 *	−0.01	−0.10 *	0.01	0.06	−0.05	−0.01	−0.10 *	−0.04	−0.01	−0.09	−0.002	0.23 *	-	-	-
BOP ^1^	−0.03	−0.01	−0.05	0.05	0.05	−0.07	−0.04	−0.03	−0.04	0.04	0.06	0.03	−0.02	0.03	−0.12	0.03	-	-
PPD ≥ 4 mm ^1^	0.10 *	−0.16 *	0.11 *	0.01	0.09	0.01	0.01	0.10 *	0.12 *	−0.02	0.07	0.07	0.06	0.12 *	0.01	−0.09	0.22 *	-
PPD ≥ 6 mm ^1^	0.14 *	−0.14 *	0.11 *	0.07	0.05	0.08	−0.02	0.02	0.08	0.06	0.06	−0.06	0.10 *	0.13 *	−0.05	−0.08	0.14 *	0.30 *

* *p* < 0.05; ^1^ 0 = absence, 1 = presence; ^2^ 0 = male, 1 = female; Abbreviation: DSWMH, deep and subcortical white matter hyperintensity; BMI, body mass index; SBP, systolic blood pressure; DBP, diastolic blood pressure; HbA1c, hemoglobin A1c; HDL, high-density lipoprotein; LDL, low-density lipoprotein; CRP, C-reactive protein; BOP, bleeding on probing; PPD, probing pocket depth.

**Table 4 ijerph-17-01694-t004:** Factors associated with DSWMH lesions: univariate logistic regression analysis.

Variables	Crude Odds Ratio	95% Confidence Interval	*p*-Value
Male (vs. female)	0.834	0.558–1.248	0.378
Age ≥ 65 years (vs. < 65 years)	3.834	2.237–6.571	<0.001
BMI ≥25 (vs. < 25)	0.902	0.600–1.354	0.618
Presence of smoking habits (vs. absence)	0.937	0.559–1.573	0.807
Presence of drinking habits (vs. absence)	1.360	0.877–2.111	0.170
Presence of regular exercise (vs. absence)	0.847	0.536–1.341	0.479
SBP ≥ 140 mmHg (vs. < 140 mmHg)	2.932	1.460–5.891	0.003
DBP ≥ 90 mmHg (vs. < 90 mmHg)	1.383	0.684–2.797	0.367
Serum parameters			
HbA1c ≥ 6.5% (vs. < 6.5%)	1.152	0.542–2.447	0.713
Triglyceride ≥ 150 mg/dL (vs. < 150 mg/dL)	0.956	0.552–1.665	0.873
HDL cholesterol <40 mg/dL (vs. ≥ 40 mg/dL)	0.769	0.345–1.714	0.521
CRP ≥ 0.3 mg/dL (vs. < 0.3 mg/dL)	0.965	0.438–2.203	0.965
Oral parameters			
Number of present teeth ≥ 20 teeth (vs. < 20 teeth)	0.458	0.154–1.361	0.160
Number of present teeth ≥ 28 teeth (vs. < 28 teeth)	0.489	0.333–0.720	<0.001
Presence of BOP (vs. absence)	0.867	0.520–1.445	0.584
Presence of PPD ≥ 4 mm (vs. absence)	1.524	1.014–2.293	0.043
Presence of PPD ≥ 6 mm (vs. absence)	2.184	1.298–3.677	0.003

Abbreviation: DSWMH, deep and subcortical white matter hyperintensity; BMI, body mass index; SBP, systolic blood pressure; DBP, diastolic blood pressure; HbA1c, hemoglobin A1c; HDL, high-density lipoprotein; LDL, low-density lipoprotein; CRP, C-reactive protein; BOP, bleeding on probing; PPD, probing pocket depth.

**Table 5 ijerph-17-01694-t005:** Factors associated with DSWMH lesions: multiple logistic regression analysis.

Variables	Adjusted Odds Ratio	95% Confidence Interval	*p*-Value
Model 1 ^1^			
Age ≥ 65 years (vs. < 65 years)	3.111	1.776–5.451	<0.001
SBP ≥ 140 mmHg (vs. < 140 mmHg)	2.578	1.256–5.288	0.010
Number of present teeth ≥ 28 teeth (vs. < 28 teeth)	0.622	0.413–0.938	0.023
Model 2 ^2^			
Age ≥ 65 years (vs. < 65 years)	2.984	1.696–5.232	<0.001
SBP ≥ 140 mmHg (vs. < 140 mmHg)	2.579	1.252–5.314	0.010
Number of present teeth ≥ 2 8 teeth (vs. < 28 teeth)	0.635	0.420–0.961	0.032
Presence of PPD ≥ 6 mm (vs. absence)	1.948	1.132–3.354	0.016

^1^ Adjusted by age, SBP, number of present teeth, and PPD ≥ 4 mm; ^2^ Adjusted by age, SBP, number of present teeth, and PPD ≥ 6 mm; Abbreviation: DSWMH, deep and subcortical white matter hyperintensity; SBP, systolic blood pressure; PPD, probing pocket depth.

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
