# Peer review of "Periodontal Condition Is Correlated with Deep and Subcortical White Matter Hyperintensity Lesions in Japanese Adults"

_ijerph, 2020, doi:10.3390/ijerph17051694_

Round 1
Reviewer 1 Report
interesting work
abstract, add one sentence explaining the 'deep and subcortical white matter hyperintensity (DSWMH) lesions '
how did you collect the sample, was it a convenient sample or a general population?, add the type of study (x-sectional), add mean (SD) for the age, for the ODD ratios, please add the 95% CI as well
introduction: very short
Talk more on 'circulating inflammatory molecules' which ones? also when you say' is also associated with the risk of numerous systemic diseases, including brain disorders' name them, should you say the 'disorders' or 'disease' do you mean for instance' Mental Disorders.' or even 'Brain Cancer', expand more
What do you mean by 'These observations indicate that periodontitis could augment brain disorders.' did you mean association, augment is not really the right word here!
expand the discussion, suggest future studies,
work on grammar and style as there are typos present
Author Response
interesting work
abstract, add one sentence explaining the 'deep and subcortical white matter hyperintensity (DSWMH) lesions '
Our response: Thank you for your suggestion. We have added a sentence to the abstract (page 1, lines 17-18).
how did you collect the sample, was it a convenient sample or a general population? add the type of study (x-sectional), add mean (SD) for the age, for the ODD ratios, please add the 95% CI as well
Our response: Thank you for your suggestions. We have revised the abstract (page 1, lines 18-20, 25-28).
introduction: very short
Our response: We have added a number of sentences that provide more background information (page 1, lines 42-43; page 2, lines 47-48; and page 2, lines 58-59).
Talk more on 'circulating inflammatory molecules' which ones? also when you say' is also associated with the risk of numerous systemic diseases, including brain disorders' name them, should you say the 'disorders' or 'disease' do you mean for instance' Mental Disorders.' or even 'Brain Cancer', expand more
Our response: Thank you for your suggestions. We have revised the relevant sections to address the reviewer’s concerns (page 1, lines 38-39). In addition, we changed the terminology from “disorders” to “diseases” for clarity (page 1, line 39; page 2, lines 49, 50 and 54; page 7, lines 168, 180, 181 and 183).
What do you mean by 'These observations indicate that periodontitis could augment brain disorders.' did you mean association, augment is not really the right word here!
Our response: Thank you for your suggestion. We have revised this sentence (page 2, line 49).
expand the discussion, suggest future studies,
Our response: Thank you for your suggestion. We have revised the Discussion (page 8, lines 229-231).
work on grammar and style as there are typos present
Our response: Thank you for your suggestion. A native English speaker has re-checked our manuscript.
Reviewer 2 Report
The manuscript entitled “Periodontal Condition Is Correlated with Deep and Subcortical White Matter Hyperintensity Lesions in Japanese Adults” requires minor revision before publication. It mainly describes that the brain disorders and periodontitis would have “positive association” then only tables were made to demonstrate several reliable observations. The findings in the research of public health is interesting. However, with the limited number of references it would be difficult to assume such feasibility. Also, the authors hypothesized that the risk of DSWMH lesions might be related to periodontitis. The assumption was made first to meet the causality of research.
Several suggestions are addressed below:
(1) The authors state that brain disorders and periodontitis would have “positive association” with a limited number of studies from references 6 to8. It would be difficult to assume such feasibility. Please include more references to support the claim. Is it possible for those brain disorders own health gums?
(2) The study enrolled 444 participants at the Asahi University Hospital in Gifu, while the title uses “Japanese adults” to represent those participants. The authors conclude the research finding in a cross-sectional study in Japan. Is it fair to say that those participants are regional or from a wider spectrum?
(3) On page 7, line 160-169: the authors point out the subtle correlation between the participants with chronic periodontitis and brain disorder. Perhaps the authors can suggest more solid numbers that can lead to such correlation.
(4) The reviewer is wondering the association of DSWMH lesions and periodontitis for the age below 65 years. Table 4 shows the correlated variable Age ≥65 years vs. <65 years. How many clinical case of DSWMH lesions for the age <65 years and <28 teeth? Please include the data in the supporting information.
(5) Please check for grammatical errors and proper punctuation use in the manuscript. The usage of English “In this study” was emphasized several times in the draft.
Author Response
The manuscript entitled “Periodontal Condition Is Correlated with Deep and Subcortical White Matter Hyperintensity Lesions in Japanese Adults” requires minor revision before publication. It mainly describes that the brain disorders and periodontitis would have “positive association” then only tables were made to demonstrate several reliable observations. The findings in the research of public health is interesting. However, with the limited number of references it would be difficult to assume such feasibility. Also, the authors hypothesized that the risk of DSWMH lesions might be related to periodontitis. The assumption was made first to meet the causality of research.
Several suggestions are addressed below:
(1) The authors state that brain disorders and periodontitis would have “positive association” with a limited number of studies from references 6 to8. It would be difficult to assume such feasibility. Please include more references to support the claim. Is it possible for those brain disorders own health gums?
Our response: Thank you for your suggestions. We have added several supporting references (ref. no. 6, 9, and 11) and revised the manuscript accordingly (page 1, lines 41-43 and page 2, lines 46-48). In addition, it is possible that patients with brain diseases can have healthy gums. In our data, the prevalence of PPD <4 mm in participants with DSWMH lesions was 26%.
(2) The study enrolled 444 participants at the Asahi University Hospital in Gifu, while the title uses “Japanese adults” to represent those participants. The authors conclude the research finding in a cross-sectional study in Japan. Is it fair to say that those participants are regional or from a wider spectrum?
Our response: The participants were regional; we have emphasized this limitation in the Discussion (page 8, line 225).
(3) On page 7, line 160-169: the authors point out the subtle correlation between the participants with chronic periodontitis and brain disorder. Perhaps the authors can suggest more solid numbers that can lead to such correlation.
Our response: Thank you for your suggestion. We have revised the manuscript accordingly (page 7, lines 169-171, 178-180).
(4) The reviewer is wondering the association of DSWMH lesions and periodontitis for the age below 65 years. Table 4 shows the correlated variable Age ≥65 years vs. <65 years. How many clinical case of DSWMH lesions for the age <65 years and <28 teeth? Please include the data in the supporting information.
Our response: Thank you for your suggestion. The prevalence of DSWMH lesions with age < 65 years and < 28 teeth was 34.9% (n=155) and 23.6% (n=104), respectively. We have added this information to the Results (page 3, lines 126-127).
(5) Please check for grammatical errors and proper punctuation use in the manuscript. The usage of English “In this study” was emphasized several times in the draft.
Our response: Thank you for your suggestion. A native English speaker has re-checked our manuscript.
Reviewer 3 Report
Periodontal Condition Is Correlated with Deep and Subcortical White Matter Hyperintensity Lesions in Japanese Adults
Summary of manuscript: This manuscript reports the relationship between deep and subcortical white matter hyperintensity (DSWMH) lesions and periodontal status in Japanese adults. This report concludes that there appears to be a positive association between DSWMH lesions and severe periodontitis in a cross-sectional study in Japan. The topic in the present study is novel and unique, and the sample size was large. However, the study design and methods of statistical analysis that you applied were inappropriate for publication as an original article.
For logistic regression analysis, the multicollinearity between dependent variables and the control of confounding factors were not considered. This seems to be a fatal mistake in statistical analysis.
The evidence that you used to divide the two groups to analyze the “number of present teeth” with cut-off value of 28 was indefinite.
The occlusal condition, including denture condition, in subjects is lack. Moreover, medication status as general condition is not considered in this study.
Author Response
Summary of manuscript: This manuscript reports the relationship between deep and subcortical white matter hyperintensity (DSWMH) lesions and periodontal status in Japanese adults. This report concludes that there appears to be a positive association between DSWMH lesions and severe periodontitis in a cross-sectional study in Japan. The topic in the present study is novel and unique, and the sample size was large. However, the study design and methods of statistical analysis that you applied were inappropriate for publication as an original article.
For logistic regression analysis, the multicollinearity between dependent variables and the control of confounding factors were not considered. This seems to be a fatal mistake in statistical analysis.
Our response: Thank you for your suggestions. Spearman's correlation analyses between each variable was conducted, and variables with |r| > 0.8 were removed to avoid multicollinearity. We have added the sentence (page 3, lines 119-120). There were no variables with |r| > 0.8 in the spearman correlation analysis in this study.
The evidence that you used to divide the two groups to analyze the “number of present teeth” with cut-off value of 28 was indefinite.
Our response: Thank you for your suggestions. Since the maximum number of present teeth, excluding wisdom teeth, is usually 28, we analyzed the number of present teeth with cut-off value of 28. We have added the sentence (page 3, lines 108-109).
The occlusal condition, including denture condition, in subjects is lack. Moreover, medication status as general condition is not considered in this study.
Our response: Thank you for your suggestions. We have emphasized these limitations in the Discussion (page 8, lines 226-228).
Round 2
Reviewer 1 Report
thank you , I don't think there is much that can be done on the this data set, this a good starting point
Author Response
thank you, I don't think there is much that can be done on this data set, this a good starting point
Our response: Thank you. We feel that the paper has been sufficiently improved by your comments.
Reviewer 3 Report
Periodontal Condition Is Correlated with Deep and Subcortical White Matter Hyperintensity Lesions in Japanese Adults
Summary of manuscript: This manuscript reports the relationship between deep and subcortical white matter hyperintensity (DSWMH) lesions and periodontal status in Japanese adults. This report concludes that there appears to be a positive association between DSWMH lesions and severe periodontitis in a cross-sectional study in Japan. The topic in the present study is novel and unique, and the sample size was large. There is however one issue that needs to be addressed.
Regarding the result of spearman's correlation analyses, author mentioned that “There were no variables with |r| > 0.8 in the spearman correlation analysis in this study.” At author response. However, this fact was not reflected in “Results” section. The result Table of spearman's correlation analyses should be added.
Author Response
This manuscript reports the relationship between deep and subcortical white matter hyperintensity (DSWMH) lesions and periodontal status in Japanese adults. This report concludes that there appears to be a positive association between DSWMH lesions and severe periodontitis in a cross-sectional study in Japan. The topic in the present study is novel and unique, and the sample size was large. There is however one issue that needs to be addressed.
Regarding the result of spearman's correlation analyses, author mentioned that “There were no variables with |r| > 0.8 in the spearman correlation analysis in this study.” At author response. However, this fact was not reflected in “Results” section. The result Table of spearman's correlation analyses should be added.
Our response: Thank you for your comments. We have added a table (Table 3) and the sentences in the Results section (page 5, lines 140-143).